# A Complementary Metal-Oxide Semiconductor (CMOS) Analog Optoelectronic Receiver with Digital Slicers for Short-Range Light Detection and Ranging (LiDAR) Systems

**DOI:** 10.3390/mi16020215

**Published:** 2025-02-13

**Authors:** Yunji Song, Sung-Min Park

**Affiliations:** 1Division of Electronic & Semiconductor Engineering, Ewha Womans University, Seoul 03760, Republic of Korea; dbswl0828@ewhain.net; 2Graduate Program in Smart Factory, Ewha Womans University, Seoul 03760, Republic of Korea

**Keywords:** APD, CTLE, differential, LA, optoelectronic, slicer, TIA

## Abstract

This paper introduces an analog differential optoelectronic receiver (ADOR) integrated with digital slicers for short-range LiDAR systems, consisting of a spatially modulated P^+^/N-well on-chip avalanche photodiode (APD), a cross-coupled differential transimpedance amplifier (CCD-TIA) with cross-coupled active loads, a continuous-time linear equalizer (CTLE), a limiting amplifier (LA), and dual digital slicers. A key feature is the integration of an additional on-chip dummy APD at the differential input node, which enables the proposed ADOR to outperform a traditional single-ended TIA in terms of common-mode noise rejection ratio. Also, the CCD-TIA utilizes cross-coupled PMOS-NMOS active loads not only to generate the symmetric output waveforms with maximized voltage swings, but also to provide wide bandwidth characteristics. The following CTLE extends the receiver bandwidth further, allowing the dual digital slicers to operate efficiently even at high sampling rates. The LA boosts the output amplitudes to suitable levels for the following slicers. Then, the inverter-based slicers with low power consumption and a small chip area produce digital outputs. The fabricated ADOR chip using a 180 nm CMOS process demonstrates a 20 dB dynamic range from 100 μA_pp_ to 1 mA_pp_, 2 Gb/s data rate with a 490 fF APD capacitance, and 22.7 mW power consumption from a 1.8 V supply.

## 1. Introduction

Short-range light detection and ranging (LiDAR) technology has emerged as a critical component in diverse fields, including autonomous navigation, robotic systems, and industrial automation [1,2,3]. As these applications demand precise object detection and mapping capabilities, there is a growing need for advanced receiver circuits capable of addressing the stringent performance requirements associated with high-speed and high-resolution data acquisition. This necessitates innovations in circuit designs to enhance signal fidelity, bandwidth, and power efficiency [4].

Optoelectronic receivers play a pivotal role in LiDAR systems by converting optical signals into electrical domains with minimal distortion [5]. Conventional single-ended transimpedance amplifiers (TIAs), while widely employed, often face challenges in rejecting common-mode noise, thereby limiting their effectiveness in high-performance scenarios. To overcome these limitations, differential receiver architectures have gained prominence due to their superior noise immunity and ability to maintain signal integrity under adverse conditions [6].

This paper presents a fully integrated analog differential optoelectronic receiver (ADOR) tailored for short-range LiDAR applications. Figure 1 illustrates the block diagram of the proposed receiver, which integrates a spatially modulated P^+^/N-well avalanche photodiode (APD) with a cross-coupled differential TIA (CCD-TIA) employing PMOS-NMOS active loads to achieve enhanced common-mode noise rejection and optimized voltage swing characteristics.

In addition, the integration of a continuous-time linear equalizer (CTLE) extends the system’s bandwidth, ensuring robust operation at elevated sampling rates. The subsequent limiting amplifier (LA) amplifies the signal for reliable digital slicing, while the inverter-based slicers deliver efficient digital outputs with minimal power consumption.

Unlike previously published receivers that exploit external off-chip photodetectors or single-ended TIA architectures [7,8], the proposed ADOR offers key advantages. First, the on-chip APD minimizes the parasitic capacitance and interconnect losses, therefore enhancing signal integrity and reducing system complexity. Second, the differential TIA structure improves noise immunity, hence mitigating the limitations of conventional single-ended TIAs. Furthermore, the compact integration of three blocks including CTLE, LA, and digital slicers enables us to realize a power-efficient solution without compromising performance.

Fabricated using a 180 nm CMOS process, the proposed receiver demonstrates a dynamic input range spanning 100 μA_pp_ to 1 mA_pp_, a maximum data rate of 2 Gb/s, and a power consumption of 22.7 mW from a 1.8 V supply. These results underscore its potential as a high-performance solution for short-range LiDAR systems, meeting the demands of modern applications that prioritize accuracy, speed, and efficiency.

## 2. Circuit Description

### 2.1. On-Chip P^+^/NW APD

On-chip APDs are developed by utilizing 180 nm CMOS technology. Figure 2 presents the cross-sectional view of the on-chip APD and its layout, which adopts the same architecture as the P^+^/N-well (NW) APD reported in [9]. AC ground is applied to the contacts of the p-substrate and NW to suppress the slow diffusion currents and enhance the operational speed. The P^+^ contact is interfaced with the TIA, while a shielded dummy APD is connected to the other input terminal of the TIA for achieving symmetry, as depicted in Figure 3. The fabricated on-chip APDs feature an octagonal geometry to mitigate premature edge breakdown [10], with the optical window having a diagonal length of 40 µm. This design results in a parasitic capacitance of 0.49 pF. The experimental data of the P^+^/NW APD indicate responsivity of 2.72 A/W at a reverse bias of 11.05 V [9].

### 2.2. Differential Transimpedance Amplifier

Figure 3 illustrates the schematic diagram of the cross-coupled differential transimpedance amplifier (CCD-TIA), where the cross-coupled active loads consisting of PMOS transistors ($M_{3}$, $M_{4}$) and NMOS source followers ($M_{5}$ with $R_{1}$, and $M_{6}$ with $R_{2}$) are exploited with their gates cross-connected to ensure symmetric differential outputs [11].

As depicted, the on-chip APD absorbs light pulses. Then, the optical signals are converted into electrical currents ($i_{pd}$), which flow toward the gate of $M_{1}$. A variable resistor ($R_{F}$) is placed between the gate (or input) and drain (or output) nodes of $M_{1}$ to provide negative feedback. Small-signal analysis reveals that the transimpedance gain of the CCD-TIA is given by the following:(1)$$\frac{v_{o}^{+}}{i_{pd}}≅\frac{{R_{F}g}_{m_{1}}}{g_{m_{1}}+g_{m_{3}}}≅R_{F}\;if\;g_{m_{1}}\gg g_{m_{3}}\;\frac{v_{o}^{-}}{i_{pd}}≅-\frac{{R_{F}g}_{m_{1}}}{g_{m_{1}}+g_{m_{3}}}≅-R_{F}\;if\;g_{m_{1}}\gg g_{m_{3}}$$
where *g_m_i__*_(*i*=1, 3, 5)_ represents the transconductance of transistor *M_i_*_(*i*=1, 3, 5)_.

The equivalent noise current spectral density of the CCD-TIA is given by the following:(2)$$\bar{I_{eq}^{2}}≅\frac{4kT}{R_{F}}+\left(\frac{4kT\Gamma g_{m_{1}}+4kT\Gamma g_{m_{3}}}{g_{m_{1}}^{2}}\right)\left(\frac{1}{R_{F}^{2}}+\omega^{2}C_{tot}^{2}\right)+\left(\frac{4kT\Gamma g_{m_{5}}+\frac{4kT}{R_{1}}}{g_{m_{5}}^{2}R_{F}^{2}}\right)\;≅\frac{4kT}{R_{F}}+\frac{4kT\Gamma (g_{m_{1}}+g_{m_{3}})\omega^{2}C_{tot}^{2}}{g_{m_{1}}^{2}}$$
where $C_{tot}$ represents the total input capacitance of $M_{1}$ including both the photodiode capacitance ($C_{pd}$) and the gate capacitance of $M_{1}$. This analysis highlights the importance of optimizing the values of $gm_{1}$ and $gm_{5}$ for superior performance, while minimizing $gm_{3}$ is critical to reduce noise contributions.

### 2.3. 3-Bit Continuous-Time Linear Equalizer (CTLE)

As shown in Figure 4, a 3-bit continuous-time linear equalizer (CTLE) is added right after the CCD-TIA to adjust the gain and bandwidth characteristics. Typically, CTLE leverages circuit techniques such as source degeneration, inductive peaking, or negative capacitance, thereby improving the gain at the Nyquist frequency with the frequency peaking phenomenon [12]. These CTLE circuits feature a simple design and high linearity, operating without requiring synchronized clock signals for equalization. In this work, a 3-bit digital-to-analog (DAC) incorporating NMOS resistors is employed for dynamic adjustment with the degeneration capacitor (C1), allowing automatic tuning of gain and bandwidth.

The transfer function of the two-stage CTLE is given by the following:(3)$$H\left(s\right)≅{\left(\frac{g_{m}R_{3}}{1+\frac{g_{m}R_{3}}{2}}\cdot \frac{1+\frac{s}{\omega_{z}}}{\left(1+\frac{s}{\omega_{p1}}\right)\cdot \left(1+\frac{s}{\omega_{p2}}\right)}\right)}^{2}$$
where $\omega_{z}=\frac{1}{R_{4}C_{1}},\omega_{p1}=\frac{1+\frac{g_{m}R_{4}}{2}}{R_{4}C_{1}},\omega_{p2}=\frac{1}{R_{3}C_{L}}$ [7].

### 2.4. Limiting Amplifier (LA) and Inverter-Base Slicers

The CTLE is followed by a limiting amplifier (LA) that consists of two-stage differential amplifiers with an active feedback circuit. Figure 5 depicts the schematic diagram of the LA, in which the active feedback circuit is integrated to enhance the bandwidth [13]. Hence, when the output of the CTLE is applied, the LA amplifies the output amplitudes to produce wide output swings. In addition, the subsequent stage of the LA, i.e., the inverter-based slicers, adjusts the CTLE output swings suitable for the proper operations in the slicer.

As the dual digital slicers, two simple inverters are exploited. Then, the output (Q) becomes high when an input is present and low when there is none. With this structure, its power consumption and chip size can be reduced. It should be noted that a slight mismatch would be intentionally introduced in these inverters so that the inevitable DC offset voltages from the previous LA could be compensated.

Figure 6 illustrates the layout of the proposed ADOR integrated with two on-chip P^+^/N-well APDs, which occupies a core area of 207 × 98 µm^2^. DC simulations reveal that the ADOR operates efficiently, consuming 22.7 mW from a 1.8 V supply.

Figure 7 shows the post-layout simulation results of the frequency response when the proposed ADOR is configured with CCD-TIA, CTLE, and LA block, respectively. First, the transimpedance gain and bandwidth of the CCD-TIA only is 53.2 dBΩ and 3.62 GHz. Then, the transimpedance gain and bandwidth of the CCD-TIA followed by the CTLE become 60.4 dBΩ and 7.7 GHz, confirming the bandwidth extension characteristic of the CTLE. Finally, the combination of the CCD-TIA, CTLE, and LA circuits yields the transimpedance gain and bandwidth of 71.1 dBΩ and 5.36 GHz.

Figure 8 presents the simulation results of the ADOR outputs for various input currents from 100 µA_pp_ to 1 mA_pp_. It is clearly seen that the output (Q) of the dual digital slicers remains stable at 1.8 V_pp_ across the whole range, thus, indicating a 20 dB input dynamic range characteristic of the ADOR.

## 3. Chip Implementation and Measurements

The ADOR chips were implemented by using a standard 180 nm CMOS technology. Figure 9 presents a chip photograph of the fabricated ADOR alongside its test setup, employing optical test instruments. The chip’s core occupies a compact area of 207 × 98 µm^2^ and operates with a DC power consumption of 22.7 mW.

Figure 10 displays the measured eye diagrams of the ADOR with a 500 μA_pp_ input current at various data rates of 500 Mb/s, 1 Gb/s, and 2 Gb/s, respectively. It is clearly seen that the ADOR provides stable output levels at each data rate.

Figure 11 presents the measured eye-diagrams for the input currents of 100 μA_pp_, 500 μA_pp_, and 1 mA_pp_ at 500 Mb/s data rate. From both Figure 11 and Figure 12, the ADOR demonstrates a 20 dB input dynamic range. This performance is acquired through the 3-bit DAC-based gain control in the CTLE, and the function of the cross-coupled active loads. The observed eye-amplitudes are constrained to 120 mV_pp_, which can be attributed to the inherent impedance mismatch between the measurement equipment and the dual digital slicers.

Specifically, the output impedance of the slicers is approximately 700 Ω, while the input impedance of the measurement equipment is 50 Ω. This mismatch results in the considerable voltage attenuation in measurements. However, it is observed that the output voltage swings remain constant, regardless of the variations in the input currents and the data rates. Additionally, the measurable SNR in Figure 11 and Figure 12 is approximately 9.5.

Figure 12 displays the measured output noise voltage of the ADOR, where the equivalent noise current spectral density is estimated to be 12.3 pA/√Hz with a transimpedance gain of 53.2 dBΩ. This corresponds to an optical sensitivity of −30.2 dBm at a BER of 10^−12^, based on the on-chip APD’s responsivity of 2.72 A/W.

Table 1 presents a performance comparison between the proposed ADOR and previous works. Ref. [8] implemented a TIA using a voltage-controlled gain amplifier (VGA), equalizer, and offset cancellation network (OCN). Although the TIA chip was implemented by using a 35 nm CMOS process, it exhibited a slow sampling rate and large power dissipation when compared to the proposed ADOR.

Ref. [14] employed an on-chip photodetector, a differential transimpedance amplifier, a limiting amplifier, and an output buffer. However, this TIA showed high input noise current density and large power dissipation despite the very high responsivity of the on-chip p-i-n photodiode.

Ref. [15] realized a transimpedance amplifier followed by a Cherry-Hooper stage with a negative impedance converter (CH with NIC) and a multi-stage post amplifier with a 50 Ω impedance matching output buffer. Yet, this design exhibited large power dissipation and an extremely large chip area.

Ref. [16] suggested a three-stage inverting amplifier with an adaptive equalizer and an AGC post amplifier with offset compensation. Compared to the proposed ADOR, it demonstrated a slow sampling rate and also large power dissipation and chip area.

Ref. [17] designed a common-source transimpedance amplifier (CS-TIA) followed by a single-to-differential amplifier and a limiting amplifier. This CS-TIA showed a slow sampling rate as well as high power dissipation and large chip area.

In contrast, this work exploits an on-chip APD with rather moderate responsivity operating at an 850 nm wavelength, and a cross-coupled differential TIA architecture to acquire higher sampling rate and lower power consumption, not to mention the small chip area. Hence, the proposed ADOR provides a low-cost low-power solution for the applications of short-range LIDAR sensors.

Although the proposed ADOR achieves a higher sampling rate compared to the prior works, its noise performance is moderately large due to its low power consumption. Therefore, the tradeoff between sensitivity and power consumption should be carefully optimized to ensure efficient operations for short-range LiDAR sensor systems where power efficiency and high-speed data acquisition are critical.

## 4. Conclusions

We have demonstrated a CMOS-based analog differential optoelectronic receiver (ADOR) optimized for short-range LiDAR sensor applications. The proposed architecture integrates key components, including a spatially modulated P^+^/N-well on-chip APD, a cross-coupled differential transimpedance amplifier (CCD-TIA), a continuous-time linear equalizer (CTLE), a limiting amplifier (LA), and inverter-based dual digital slicers. The integration of an additional dummy APD at the differential input node enhances the common-mode noise rejection, outperforming traditional single-ended TIAs.

The ADOR achieves a wide bandwidth with efficient performance, providing a dynamic range of 20 dB from 100 μ$A_{pp}$ to 1 m$A_{pp}$, a sampling rate of 2 Gb/s, and low power consumption of 22.7 mW at dual 1.8 V supply. The design demonstrates the potential for high-performance ADCs suitable for LiDAR systems, offering both high-speed operation and energy efficiency. Additionally, the adoption of on-chip APDs reduces system packaging costs, making this approach a viable solution for cost-effective, integrated LiDAR sensor systems. The proposed architecture exemplifies the benefits of fully differential designs in optimizing signal fidelity and power efficiency for short-range LiDAR applications.

Future research will focus not only on enhancing sensitivity but also on further optimizations of the CTLE and LA design to reduce power dissipation, thereby enhancing the efficiency of the proposed ADOR for short-range LiDAR sensor applications.

## Figures and Tables

**Figure 1 micromachines-16-00215-f001:**
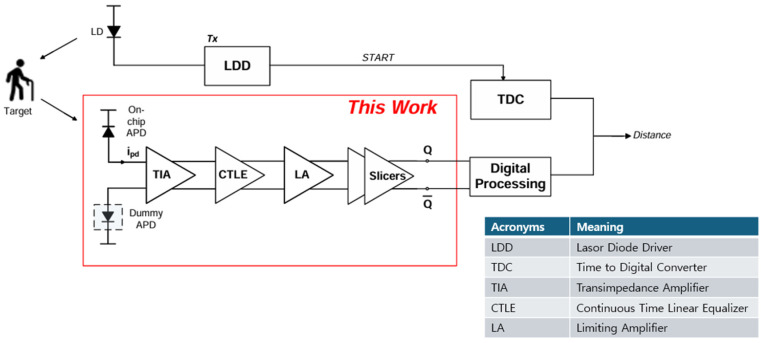
Block diagrams of the proposed ADOR.

**Figure 2 micromachines-16-00215-f002:**
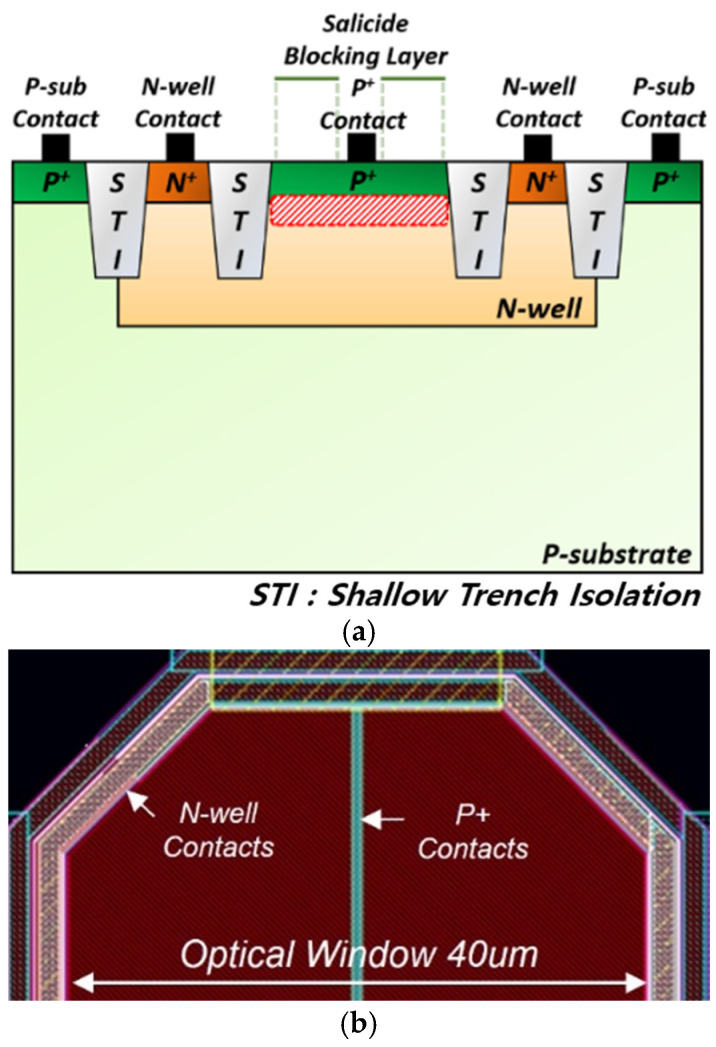
(**a**) Cross-sectional view of the P^+^/N-well APD integrated on-chip; (**b**) its layout.

**Figure 3 micromachines-16-00215-f003:**
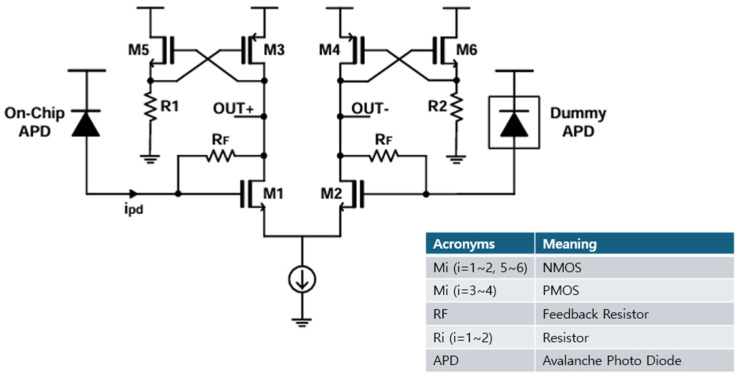
Schematic diagram of the CCD-TIA.

**Figure 4 micromachines-16-00215-f004:**
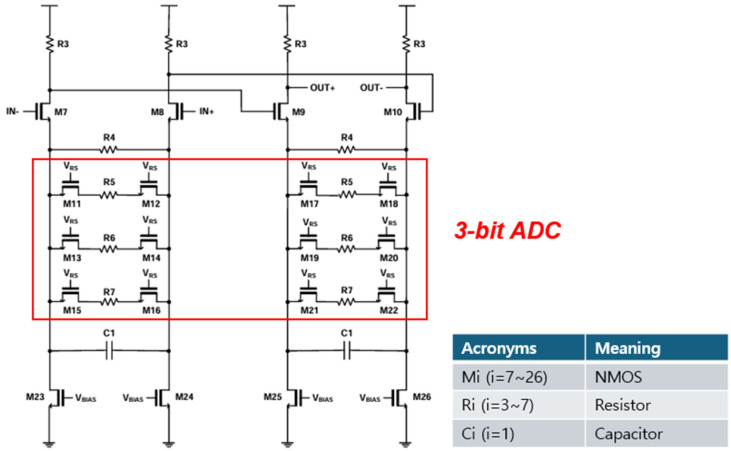
Schematic diagram of the 3-bit CTLE.

**Figure 5 micromachines-16-00215-f005:**
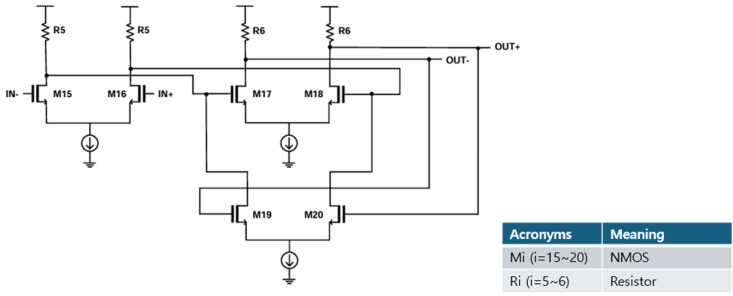
Schematic diagram of the LA.

**Figure 6 micromachines-16-00215-f006:**
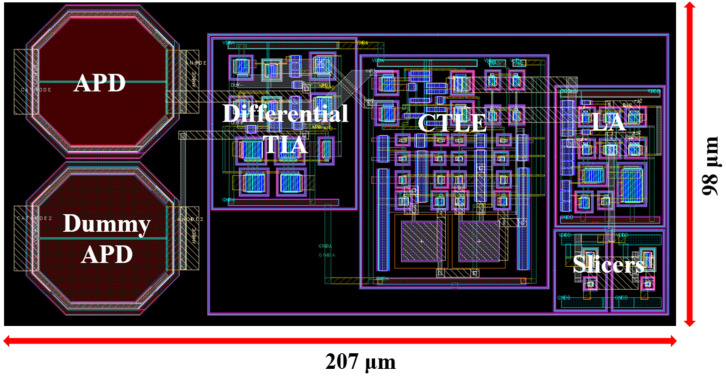
Layout of the proposed ADOR.

**Figure 7 micromachines-16-00215-f007:**
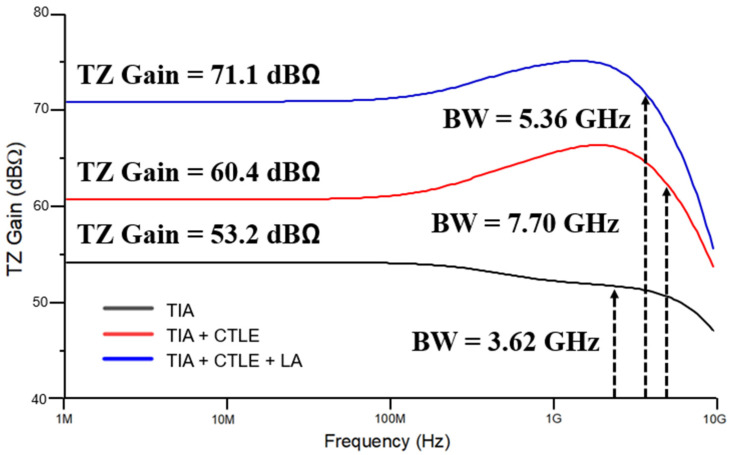
Simulated frequency responses of the CCD-TIA, CTLE, and LA circuits.

**Figure 8 micromachines-16-00215-f008:**
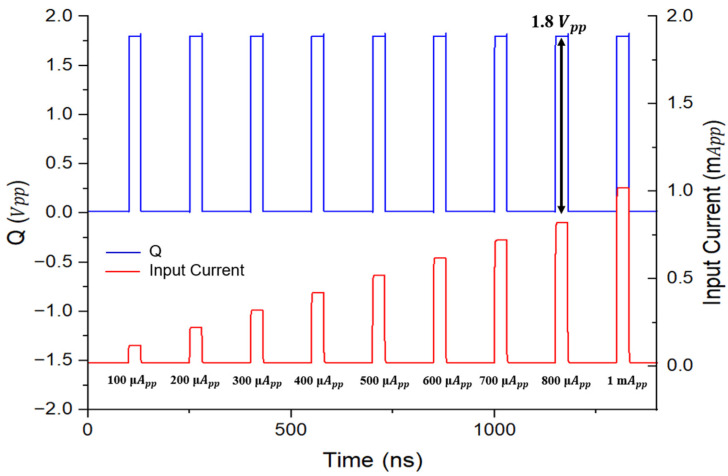
Simulated output waveforms of the ADOR corresponding to varying input current levels.

**Figure 9 micromachines-16-00215-f009:**
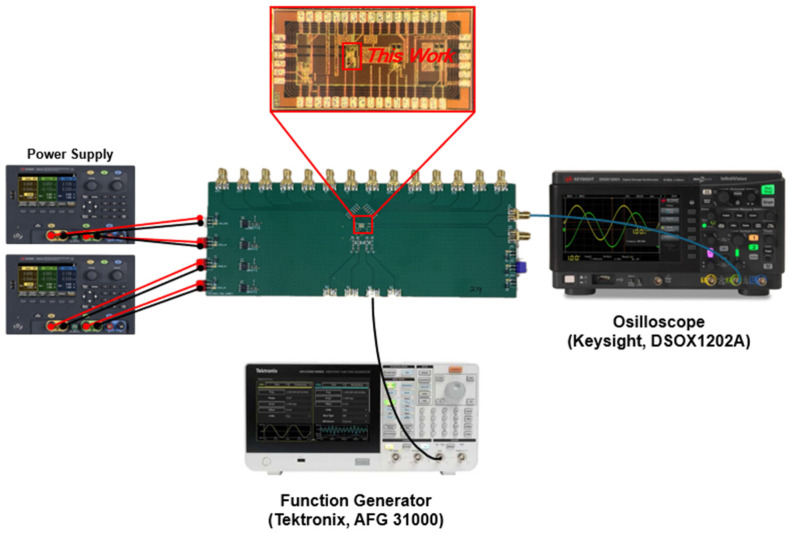
Photograph of the fabricated ADOR chip and its corresponding test setup (inc. optical test).

**Figure 10 micromachines-16-00215-f010:**
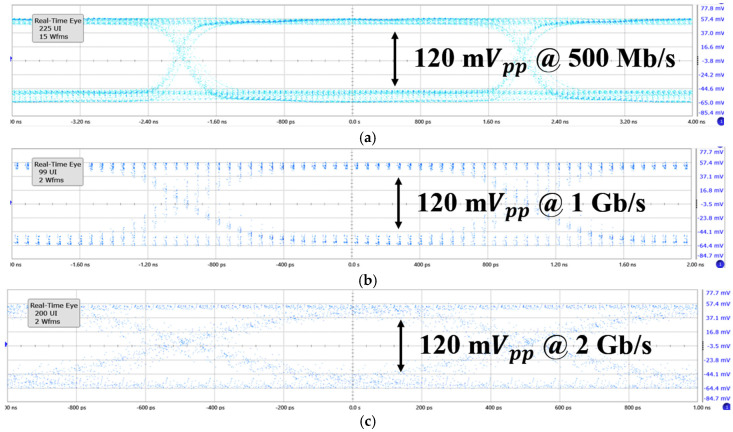
Measured eye diagrams of the ADOR with a 500 μA_pp_ input current at various data rates of (**a**) 500 Mb/s, (**b**) 1 Gb/s, and (**c**) 2 Gb/s, respectively.

**Figure 11 micromachines-16-00215-f011:**
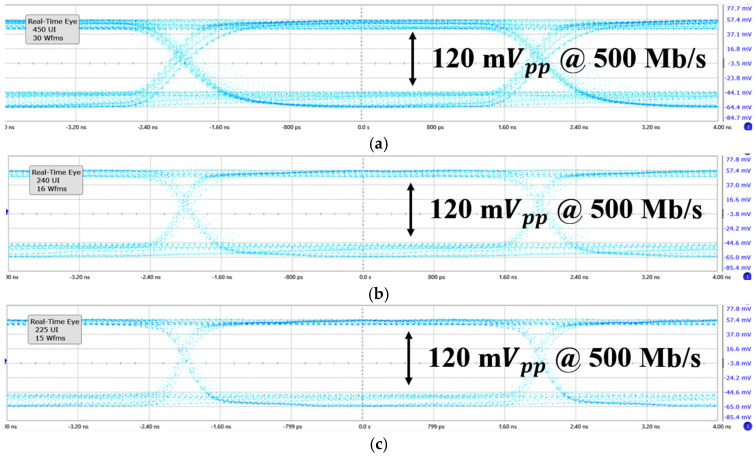
Measured eye-diagrams of the ADOR at 500-M/bs for various input currents of (**a**) 100 μA_pp_, (**b**) 500 μA_pp_, and (**c**) 1 mA_pp_, respectively.

**Figure 12 micromachines-16-00215-f012:**
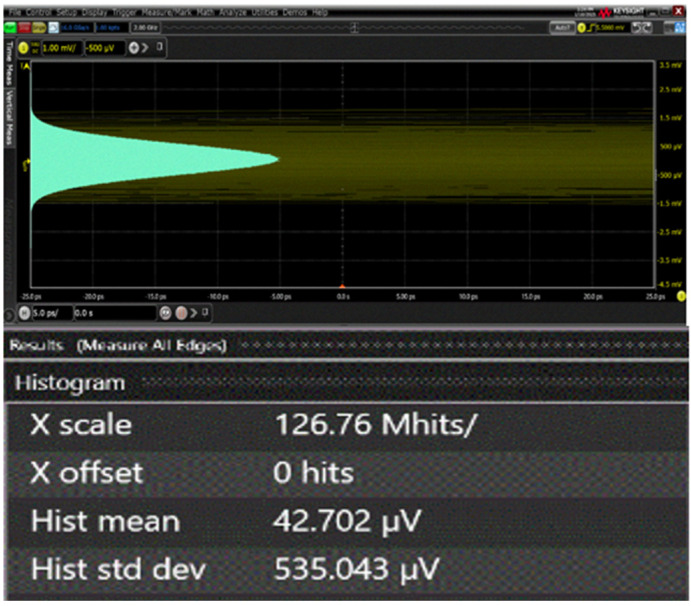
Measured output noise voltage of the ADOR.

**Table 1 micromachines-16-00215-t001:** Performance comparison of the proposed ADOR with the recently published TIAs.

Parameters		[8]	[14]	[15]	[16]	[17]	This work
CMOS technology (nm)		35	180	180	180	180	180
PD	Type	Off-chip *	On-chip (p-i-n PD)	Off-chip *	Off-chip *	Off-chip *	On-chip (APD)
	C_pd_ (pF)	0.65	0.416	N/A	3	N/A	0.5
	Responsivity (A/W)	0.53	20	0.03	0.44	N/A	2.72
	Wavelength (nm)	750	850	850	650	N/A	850
Input configuration		VGA	Diff.	Diff.	SF ^‡^	CS ^‡^	Diff.
Gain control		No	No	No	Yes	No	Yes
Sampling rate (Gb/s)		1.8	2	1.2	1.25	1.25	2
Input noise current density (pA/$\sqrt{Hz}$)		5	23.6	N/A	N/A	12	12.3
Dynamic range (dB)		N/A	N/A	41	N/A	N/A	20
Power dissipation per channel (mW)		214.5 @3.3 V	50 @ 3.3 V	450 @ 1.5 V	110 @ 1.8 V	68 @ 1.8 V	22.7
Chip area (mm^2^)		0.8	N/A	4.5	0.09	0.35	0.02

^‡^ single-ended, * equivalent circuit model of PD.

## Data Availability

Data are contained within the article.

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
