# Peer review of "A Complementary Metal-Oxide Semiconductor (CMOS) Analog Optoelectronic Receiver with Digital Slicers for Short-Range Light Detection and Ranging (LiDAR) Systems"

_micromachines, 2025, doi:10.3390/mi16020215_

Round 1

Reviewer 1 Report

Comments and Suggestions for Authors

This manuscript introduces an analog differential optoelectronic receiver for short-range LiDAR system. Please find my comments below: 

1. The novelty of this work is not clear. This work is similar to [11] and [13]. Can you highlight the main novel contributions?

2. The CCD-TIA circuit in Section 2.B is missing citations? For Section 2.A, elaborate on the design and simulation of the ADP.

3. The CTLE schematic diagram in Figure 4 is incomplete, missing the 3-bit DAC circuit. 

4. Equation (3) and Figure 4 don't match (R3=RL and R4=Rs ?). 

5. The description of the LA circuit in Section 3.C is confusing. Please re-write this subsection and provide a general block digram then show the single gain-cell. 

6. The measurements in Section 3 are not comprehensive. Can you add the measured frequency response of the different circuit in the proposed receiver? Measured BER as a function of average optical power at different data rates? 

7. Can you add post-layout simulation results for the CCD-TIA, CTLE, and LA circuits.

Reviewer 2 Report

Comments and Suggestions for Authors

Review for micromachines (Manuscript ID: #3476033)

Title: “A CMOS Analog Optoelectronic Receiver with Digital Slicers for Short-Range LiDAR Systems”.

Authors: Yunji Song , Sung-Min Park

Comments:

This work investigates a CMOS Analog Optoelectronic Receiver with Digital Slicers for Short-Range LiDAR Systems.

The paper is novel, well-presented, and well-organized. The manuscript is very well-written, with comprehensive experiments and analyses. The reported results from the fabricated ADOR chip, implemented using a 180-nm CMOS process, demonstrate a 20 dB dynamic range from 100 μApp to 1 mApp, a 2 Gb/s data rate with a 490 fF APD capacitance, and 22.7 mW power consumption from a 1.8 V supply—these results are remarkable. This paper makes a meaningful contribution to the field of optoelectronic receiver design for LiDAR systems. I recommend this manuscript for publication, pending possible minor revisions.

Additionally, I have the following minor (optional) comments for the authors:

1. The comparative analysis (Table 1) effectively positions the proposed ADOR against prior works. The table highlights key advantages, such as lower power consumption, a higher sampling rate, and a compact chip area. The discussion surrounding these comparisons is insightful, though a deeper exploration of trade-offs in sensitivity and noise performance would be beneficial.

2. A brief mention of future research directions, such as potential enhancements in responsivity or power efficiency, would add value.

3. A more explicit discussion on trade-offs and design constraints would further strengthen the manuscript.

Overall, this work is highly interesting and significant. The presented results are impressive. I recommend the publication of this work, subject to minor revisions as outlined above.

End

Author Response

1. The comparative analysis (Table 1) effectively positions the proposed ADOR against prior works. The table highlights key advantages, such as lower power consumption, a higher sampling rate, and a compact chip area. The discussion surrounding these comparisons is insightful, though a deeper exploration of trade-offs in sensitivity and noise performance would be beneficial.

(ans.) Thank you for your feedback. We have included the discussion related to Table 1 in the revised manuscript as below (line 222-226).

Although the proposed ADOR achieves a higher sampling rate compared to the prior works, its noise performance is moderately large due to its low power consumption. Therefore, the tradeoff between sensitivity and power consumption should be carefully optimized to ensure efficient operations for short-range LiDAR sensor systems where power efficiency and high-speed data acquisition are critical.”

2. A brief mention of future research directions, such as potential enhancements in responsivity or power efficiency, would add value.

(ans.) Thank you for your feedback. We have added the future directions in conclusion as follows (line 246-248):

“Future research will focus not only on enhancing sensitivity, but also on further optimizations of the CTLE and LA design to reduce power dissipation, thereby enhancing the efficiency of the proposed ADOR for short-range LiDAR sensor applications.”

3. A more explicit discussion on trade-offs and design constraints would further strengthen the manuscript.

(ans.) We have included the discussion on tradeoffs and constraints, as described in the answer #1.

Reviewer 3 Report

Comments and Suggestions for Authors

Dear Authors,

I have a few suggestions for your consideration to improve the manuscript's readability.

1. Figure 1 is shown but not referenced or discussed in the text.

2. The caption of Figure 1 needs to define all the acronyms used in the figure.  It is better to make the figure self-explanatory without having to refer back to the text for the definitions of the acronyms used.

3.  I would also recommend expanding the Introduction to discuss the proposed receiver concept's differences and advantages in comparison with other published receiver demonstrations for LiDAR applications.

4.  What is "STI" used in Figure 2?  Please define in caption.

5.  I would recommend inclusion of definitions of variables and acronyms used in Figures 3, 4 and 5.

6.  In Figure 8, the current values are a bit overcrowded.  Is there a better way of displaying them in a way what is more readable?

Best regards.

Author Response

1. Figure 1 is shown but not referenced or discussed in the text.

(ans.) We have added discussion of Figure 1 in the manuscript as below (line 41-45):

“Figure 1 illustrates the block diagram of the proposed receiver, which integrates a spatially modulated P+/N-well avalanche photodiode (APD) with a cross-coupled differential TIA (CCD-TIA) employing PMOS-NMOS active loads to achieve enhanced common-mode noise rejection and optimized voltage swing characteristics.”

2. The caption of Figure 1 needs to define all the acronyms used in the figure.  It is better to make the figure self-explanatory without having to refer back to the text for the definitions of the acronyms used.

(ans.) Thank you for your feedback. We have indicated the abbreviations in Figure 1.

3. I would also recommend expanding the Introduction to discuss the proposed receiver concept's differences and advantages in comparison with other published receiver demonstrations for LiDAR applications.

(ans.) Thank you for your feedback. We have revised the content of the introduction as follows (line 50-56):

“Unlike previously published receivers that exploit external off-chip photodetectors or single-ended TIA architectures, the proposed ADOR offers key advantages. First, the on-chip APD minimizes the parasitic capacitance and interconnect losses, therefore enhancing signal integrity and reducing system complexity. Second, the differential TIA structure improves noise immunity, hence mitigating the limitations of conventional single-ended TIAs. Furthermore, the compact integration of three blocks including CTLE, LA, and digital slicers enables to realize a power-efficient solution without compromising performance.”

4. What is "STI" used in Figure 2?  Please define in caption.

(ans.) STI stands for ‘Shallow Trench Isolation’ functioning as a guard ring to boost avalanche gain along with a compact N-well depletion region that supports broader bandwidth. We have added this to Figure 2.

5. I would recommend inclusion of definitions of variables and acronyms used in Figures 3, 4 and 5.

(ans.) We have revised Figure 3, Figure 4, and Figure 5.

6. In Figure 8, the current values are a bit overcrowded.  Is there a better way of displaying them in a way what is more readable?

(ans.) Thanks a lot for this comment. In Figure 8, we tried to show that the output waveforms of Q are constant with the large variations of the input currents from 100 μApp to 1 mApp. Therefore, we just have reduced the size of the words (numbers) in Figure 8.

Round 2

Reviewer 1 Report

Comments and Suggestions for Authors

The authors have addressed all my comments.